# A Frequency Estimation Scheme Based on Gaussian Average Filtering Decomposition and Hilbert Transform: With Estimation of Respiratory Rate as an Example

**DOI:** 10.3390/s23083785

**Published:** 2023-04-07

**Authors:** Yue-Der Lin, Yong-Kok Tan, Tienhsiung Ku, Baofeng Tian

**Affiliations:** 1Department of Automatic Control Engineering, Feng Chia University, Taichung 40724, Taiwan; 2Ph.D. Program of Electrical and Communications Engineering, Feng Chia University, Taichung 40724, Taiwan; 3Department of Anesthesiology, Changhua Christian Hospital, Changhua 50051, Taiwan; 4Artificial Intelligence Development Center, Changhua Christian Hospital, Changhua 50051, Taiwan; 5College of Instrumentation and Electrical Engineering, Jilin University, Changchun 130061, China

**Keywords:** empirical mode decomposition (EMD), ensemble empirical mode decomposition (EEMD), Gaussian average filtering decomposition (GAFD), Hilbert–Gauss transform (HGT), Hilbert–Huang transform (HHT), photoplethysmography (PPG), respiratory rate (RR), seismocardiogram (SCG), time-frequency analysis (TFA)

## Abstract

Frequency estimation plays a critical role in vital sign monitoring. Methods based on Fourier transform and eigen-analysis are commonly adopted techniques for frequency estimation. Because of the nonstationary and time-varying characteristics of physiological processes, time-frequency analysis (TFA) is a feasible way to perform biomedical signal analysis. Among miscellaneous approaches, Hilbert–Huang transform (HHT) has been demonstrated to be a potential tool in biomedical applications. However, the problems of mode mixing, unnecessary redundant decomposition and boundary effect are the common deficits that occur during the procedure of empirical mode decomposition (EMD) or ensemble empirical mode decomposition (EEMD). The Gaussian average filtering decomposition (GAFD) technique has been shown to be appropriate in several biomedical scenarios and can be an alternative to EMD and EEMD. This research proposes the combination of GAFD and Hilbert transform that is termed the Hilbert–Gauss transform (HGT) to overcome the conventional drawbacks of HHT in TFA and frequency estimation. This new method is verified to be effective for the estimation of respiratory rate (RR) in finger photoplethysmography (PPG), wrist PPG and seismocardiogram (SCG). Compared with the ground truth values, the estimated RRs are evaluated to be of excellent reliability by intraclass correlation coefficient (ICC) and to be of high agreement by Bland–Altman analysis.

## 1. Introduction

Abnormal respiratory rate (RR) has been shown to be a sensitive indicator of acute respiratory dysfunction [1]. Increased RR is also an index for the elderly patients of consecutive acute admissions to a geriatric unit before clinical diagnosis [2]. Using elevated RR as a sign for therapeutic interventions in internal medicine and emergency departments is reported to be effective in reducing the incidence of subsequent cardiopulmonary arrest and the associated morbidity and mortality [3,4]. The changes in RR can also serve as a leading marker of SARS-CoV-2 infections [5]. In addition, the variation in RR is also implicated in breathing disorders (such as sleep apnea, pneumonia, and dyspnea), emotional stress, pain-induced psycho-behavioral changes, cognitive tasks, environment stress and physical effort during sport [6]. All observations indicate that the rate of respiration is a critical vital sign, not only in clinical medicine, but also in exercise. In practical settings, respiration signal can be acquired by transthoracic impedance plethysmography [7], electrocardiogram (ECG) [8,9], capnometry [10], temperature sensors [11,12], photoplethysmography (PPG) [13], wearable strain sensor [14], seismocardiogram (SCG) [15] and stretchable conductive fabric [16]. In addition to sensing technologies [17,18], the signal processing algorithm is another important issue for the estimation of RR [17].

Frequency estimation technique plays a crucial role in RR estimation [19]. The method of frequency estimation can be implemented by discrete Fourier transform (DFT) [20], eigen-analysis approach [21], wavelet-based scalogram [22] and Hilbert–Huang transform (HHT) [23,24]. Among these methods, the conventional Fourier-based techniques are constrained by a limited frequency resolution [24]. Modern parametric spectral estimation, such as autoregressive (AR) modeling, has the advantages of shorter data length required in computation and has a higher frequency resolution when compared with Fourier-based methods [25]. However, high signal-to-noise ratio (SNR) and proper selection on model order are required for AR modeling. The signal usually needs preprocessing, and the optimal order selection method, such as Akaike information criterion (AIC), criterion autoregressive transfer function (CAT) or final prediction error (FPE), is generally needed in AR spectral estimation [25]. Taking these issues into consideration, Fleming and Tarassenko proposed an algorithm based on AR modeling that can estimate RR from a PPG signal, with a mean error of 0.04 breaths per minute (bpm) [26]. Because respiration is a time-varying signal, analysis of time-frequency domain is a feasible approach for RR estimation. To attain this goal, Chon, et al. developed an algorithm termed the variable-frequency complex demodulation (VFCDM) to estimate RR from PPG signals [27]. VFCDM can provide superior performance when compared with AR modeling and continuous wavelet transform (CWT) for RR estimation. However, the VFCDM algorithm requires at least one-minute length of signal for the estimation, and this may lead to a heavier computation load and a constraint on real-time tracking the time-varying changes of RR. To reduce the computational complexity, a one wavelet-based embedded algorithm has been proposed to estimate RR from a PPG signal, and the mean square error (MSE) is shown to be as small as 0.2534 bpm (compared with the ground truth derived from a respiration signal) [13]. Another algorithm based on wavelet decomposition (by db6 wavelet) and complex Morlet wavelet has been developed to estimate RR from SCG under resting state [15]. Although RR can be estimated in PPG or in SCG by wavelet-based algorithms, there is still a lack of one unified approach for RR estimation in different scenarios, since the wavelet kernel adopted in [13,15] are different in essence. 

The strict request on biomedical signal analysis arises from the nonstationary characteristics and the nonlinearity underlying the physiological process. HHT is a two-step procedure for data analysis. Its first step is to derive the so-called intrinsic mode function (IMF) by empirical mode decomposition (EMD) [28] or ensemble empirical mode decomposition (EEMD) [29]. Their common advantage is that the bases are data adaptive, which means the decomposition can be conducted without the need for any a priori knowledge on the data. This property makes such approaches very feasible for nonstationary and nonlinear signal analysis. The second step of HHT is to take Hilbert transform on the decomposed IMFs. Because IMFs are separated in nearly pure mode, their corresponding instantaneous frequencies can be highlighted in the Hilbert spectrum with high frequency resolution. Because of the fascinating characteristics mentioned above, HHT has been demonstrated to be powerful and effective in engineering, financial and geophysical data [30]. To date, HHT has also been widely adopted in biomedical signal analysis [31,32,33]. However, drawbacks still exist for EMD or EEMD. The widely mentioned ones include mode mixing, unnecessary redundant decomposition and boundary effect [34]. There is still room for further improvement of EMD-like methods.

To overcome the inherent deficits of EMD and EEMD, Lin et al. proposed one algorithm that is termed the Gaussian average filtering decomposition (GAFD) and has demonstrated that the respiration component can be separated reliably from finger PPG, wrist PPG and SCG, even in the case of changed breathing rate during signal measurement [35]. This novel algorithm adopts a procedure that is similar to EMD [28], but the IMFs are sifted iteratively by Gaussian average filters. In the performance comparison, GAFD has been verified to be superior to EMD and EEMD in the mentioned scenarios [35]. GAFD can sift the respiration component from PPG or SCG, but only a signal pattern is acquired in the result. In addition, the respiration pattern may appear at the distinct IMF in different scenarios. To estimate RR, this paper further applies Hilbert transform on the decomposed IMFs that are separated by GAFD to overcome the conventional deficits of EMD and EEMD, and in the meantime, can maintain the advantage of high-frequency resolution in frequency estimation. With the combination of GAFD and Hilbert transform, we hereafter name this novel method the Hilbert–Gauss transform (HGT). The advantage of Hilbert spectrum from the decomposed IMFs is the estimation precision on instantaneous frequency, even under the nonstationary cases. Because a tiny difference exists between the respiratory waveforms of inhalation and exhalation, this may cause the estimated instantaneous frequency varying with respiration, even when the subject is kept at a controlled breathing pace. Because the human respiration frequency is within a specified range [36], the proposed method firstly screens the potential respiratory IMF from the mean instantaneous frequency of the derived Hilbert spectrum. The mean instantaneous frequency is then used to estimate RR. This new method is shown to be effective for RR estimation in finger PPG, wrist PPG and SCG. Taking the comparison with the ground truth RR estimated from respiration signal, the estimated RR by the proposed method has been verified to be of excellent reliability by intraclass correlation coefficient (ICC) and be of high agreement by Bland–Altman analysis.

The rest of this paper is organized as follows. Section 2 primarily contains the data resources utilized in this study and the proposed method for RR estimation. The experimental results and the related discussion are shown in Section 3. Finally, the conclusions are given in Section 4.

## 2. Materials and Methods

This section primarily contains two parts, one presents the materials adopted in this study and the other presents the introduction for the proposed method. In the *Materials* section, the databases and conducted experiments for signal collection are covered firstly, and the computing resources (including software and computing environment) utilized in this research are subsequently introduced. The proposed method is composed of HGT and RR estimation algorithms. They are introduced separately in their specific subsections. To verify the feasibility of the proposed algorithm, the derived results are compared with those acquired from the respiration signal and are tested by intraclass correlation coefficient (ICC) [37,38] and Bland–Altman agreement analysis [39], respectively, which are introduced in the last part of *Methods* section. 

### 2.1. Materials

#### 2.1.1. Data Resources and Conducted Experiments

In this study, finger PPG, wrist PPG and SCG with synchronously recorded respiration signal are used to verify the feasibility of the proposed algorithm. In which, the finger PPG signals are selected from the MIMIC (Multi-parameter Intelligent Monitoring for Intensive Care) database [40] of PhysioNet [41]. In this database, all signals were sampled at 500 Hz and the data were collected in the intensive care unit (ICU) of Boston’s Beth Israel Hospital. As shown on the website of the MIMIC database (https://physionet.org/content/mimicdb/1.0.0/ (accessed on 1 February 2023)), there are a total of 72 records, but only 52 records contain both respiration and PPG signals. Because the PPG signals in records 418 and 427 are almost flat for the entire duration of their measurement, only 50 records are utilized in this research. Due to the limited linear range of bedside monitor, the signals recorded in the ICU may contain pattern saturation. The interruption or abrupt change may also appear in the signals because of sensor detachment, medical manipulation or patient movement. The online data viewing tool LightWAVE was utilized to sift the qualified segments from the 50 records [42]. Only segments of good quality (no saturation, no signal interruption and no interference in the signals) are selected for computer experiments. Finally, a total amount of 1000 data is included in the finger PPG database. Each data point consists of synchronous respiration and PPG signals with a fixed length of 60 s.

In this research, the wrist PPG signals were measured by the green light PPG module of ASUS VivoWatch™ through the toolkit provided by ASUSTek Computer Inc., Taiwan. The wrist PPG acquired through the toolkit was connected to the signal processing breadboard SS39L (BIOPAC^®^ Systems, Inc., Goleta, CA, USA). A dedicated cable SS60L (BIOPAC^®^ Systems, Inc., Goleta, CA, USA) for SS39L was used to convey the signals to the multifunction physiological data acquisition (DAQ) system MP30 (BIOPAC^®^ Systems, Inc., Goleta, CA, USA). The respiration signal was measured synchronously by the respiration belt transducer SS5LB (BIOPAC^®^ Systems, Inc., Goleta, CA, USA). Both signals were simultaneously acquired by Biopac Student Lab Pro analysis software with band-pass filtering of 0.05 to 35 Hz and the sampling frequency was 100 Hz for each signal. Five healthy subjects (3 males and 2 females, aged 23 ± 1 years) without a history of cardiopulmonary diseases were recruited from Feng Chia university. The conducted experiments were approved by the Institutional Review Board of Changhua Christian Hospital, Taiwan. The wrist PPG signal has been shown to be easily modulated by respiration [35]. To investigate the influence of respiration on the PPG pattern, the two-minute experiments were conducted five times for each subject. During the experiments, the subjects were requested to control their breathing speed at one fixed frequency in the first minute and change to another speed in the second minute. The controlled respiration frequencies range from 0.1 Hz (6 bpm) to 0.25 Hz (15 bpm), which were randomly selected in each experiment. In summary, there are a total of 25 data points collected in the wrist PPG database, and each signal is of 2 min length. 

In this paper, the SCG signals are selected from the CEBS (Combined measurement of ECG, Breathing and Seismocardiograms) database [43] of PhysioNet [41]. This database was established by the Electronic and Biomedical Instrumentation research group of Polytechnic University of Catalonia (UPC, Barcelona, Catalonia, Spain) with ECG, breathing signal and SCG being measured simultaneously. Twenty healthy volunteers (twelve males and eight females, aged between 19 and 30) were recruited for the experiments. Each subject was requested to keep quiet and awake in supine position on the bed and the data were recorded from the basal state (5 min), period of listening to the classic music (50 min) till the post-resting state (5 min). There are four channels of signals for each record. Channels 1 and 2 are ECG signals (lead I and II, bandwidth of 0.05 Hz to 150 Hz), channel 3 is the respiration signal (bandwidth between 0.05 Hz and 10 Hz, by SS5LB piezoresistive sensor, BIOPAC^®^ Systems, Inc., Goleta, CA, USA) and channel 4 is SCG (bandwidth of 0.5 Hz to 100 Hz, by LIS344ALH, STMicroelectronics). The data were collected by the multifunction physiological data acquisition (DAQ) system MP36 (BIOPAC^®^ Systems, Inc., Goleta, CA, USA) and the sampling frequency is 5000 Hz for each signal in the original database. With the help of online data viewing tool LightWAVE [42], it can be observed that the respiration signals in some records (such as 006, 007, 009, 010, 015, 016 and 018) are not qualified enough. The signals may contain artefacts (perhaps due to motion or other unknown reasons) or very irregular patterns, rendering the rate of respiration difficult to identify, even by visual inspection. The criterion for data selection is that the respiration signal must be regular for a one-minute duration and the rate of respiration must be able to be clearly determined by visual inspection during this interval. Finally, 500 segments of one-minute length screened from 13 subjects were collected in the SCG database. 

The signal selection and the experiments were conducted under the supervision of a clinical expert on our team to assure the quality of signals collected. As this research uses a database composed of three different types of datasets, Table 1 provides an overview of important database details, such as the number of participants, age range of the subjects, experimental conditions and the number of records in each dataset. 

The sampling frequency may influence the computation speed and the frequency resolution in frequency estimation. Because the sampling frequencies in the above-mentioned databases are different, the sampling frequencies for finger PPG, wrist PPG and SCG have been unified to be 100 Hz by resampling (using the resample function of MATLAB, MathWorks Inc., Natick, MA, USA) before further analysis.

#### 2.1.2. Computing Resources

In this study, the computer experiments are conducted on the MacBook Pro (Apple M1 Max, with 64 GB memory size, Apple Inc., Cupertino, CA, USA). The codes for GAFD and Bland–Altman analysis are developed by Python (version 3.9.12), with installed libraries NumPy (version 1.21.5), SciPy (version 1.7.3) and Matplotlib (3.5.3). The Python code of GAFD can be found on GitHub [44]. In the estimation of RR, the EMD toolbox in Python [45] is required for Hilbert transform. The code for Bland–Altman analysis in Python version can be downloaded from GitHub [46]. In addition, PyWavelets package [47] is utilized to estimate RR from respiration signal by complex Morlet wavelet, where the sample code can be found on GitHub [48]. The MATLAB code shared by Salarian [49] is utilized for the computation of ICC.

### 2.2. Methods

#### 2.2.1. Hilbert–Gaussian Transform (HGT)

Similar to HHT, HGT is also a two-step computing procedure. Its first step is GAFD, which is, in its essence, a low-pass filtering procedure conducted by using the Gaussian window. Let the signal to be analyzed be denoted as s[n], for 0≤n≤N−1. The discrete Gaussian window of length (2M+1) is given by:(1)$$w[m]=e^{-{(\alpha \cdot m/M)}^{2}/2},\;\mathrm{for}\;-M\;\leq \;m\;\leq \;M,$$
where α is the parameter inversely proportional to the standard deviation of the Gaussian distribution. The value of α is suggested to be at least 2.45, such that values at the end points of the window are less than 5% of the maximal window value [35]. In this study, α is selected to be 4.0728, which makes the end values as small as 0.025% of the maximal window value. In this situation, the characteristics of continuous Gaussian function can be approximately preserved in its discrete version. One useful characteristic is its corresponding spectrum is also of Gaussian shape [50]. In practical application, the normalized Gaussian window is utilized for computation, such that the energy can be kept unchanged during the average filtering procedure. The normalized Gaussian window can be obtained as follows:(2)$$w_{G}\left[m\right]=w[m]/\left(\sum_{l=-M}^{M}w[l]\right),\;\mathrm{for}\;-M\leq m\leq M.$$

Because signal s[n] is only defined in the interval 0≤n≤N−1, the signal needs to be extended outside the boundaries before the filtering operation with $w_{G}[\cdot ]$. There are four types of extension style in the developed Python code [44], they are “constant”, “periodical”, “reflection” and “double-symmetrical reflection”, respectively. The detailed description may refer to [35]. This paper adopts “double-symmetrical reflection” for all computer experiments because this extension style can effectively eliminate the boundary effect during the decomposition procedure based on the experimental findings. Let the extended result of signal s[⋅] over both boundary points be represented by $s_{e}[n],$ and the interval of n now becomes −*M* ≤ *n* ≤ *N* + *M* − 1 after the extension. In GAFD, the instantaneous mean is acquired by the following moving-average operation
(3)$$m_{i}\left[n\right]=\sum_{m=-M}^{M}w_{G}\left[m\right]\cdot s_{e}[m+n],\;\mathrm{for}\;0\;\leq \;n\;\leq \;N-1.\;$$

Because the Gaussian window $w_{G}[\cdot ]$ is of symmetrical shape, the computation of instantaneous mean in Equation (3) is, in its essence, the convolution sum of $w_{G}[\cdot ]$ and $s_{e}[\cdot ]$. In frequency domain, it is the direct multiplication between the spectra of $w_{G}[\cdot ]$ and $s_{e}[\cdot ]$. As the spectrum of Gaussian window is symmetric Gaussian shape centered at 0, this implies that the moving-average operation of Equation (3) is inherently a low-pass filtering on $s_{e}[\cdot ]$. After the instantaneous mean $m_{i}[\cdot ]$ has been acquired, the decomposed IMF can be derived by: (4)$$r[n]=s[n]-m_{i}[n],\;\mathrm{for}\;0\;\leq \;n\;\leq \;N-1.$$

The decomposition procedure is then conducted on r[⋅] until any stop criterion has been reached, which is similar to EMD [28]. There are three stop criteria in in the developed Python code [44]. The first one is the energy ratio of the original signal to the residual after the i-th IMF decomposition. The decomposition will terminate if the energy ratio is greater than the threshold. Another criterion is the energy difference between the neighboring IMF. If the energy difference is less than the dedicated threshold, the decomposition will also be stopped. The last criterion will be introduced later, which is related to the length of Gaussian window. 

One important issue for the filtering operation is the selection of window length for $w_{G}\left[\cdot \right]$. As mentioned in [35], there are several approaches can be selected to attain the goal. In this research, the method proposed by Lin et al. [51] is adopted and the value of M is obtained by:(5)$$\;M=2\cdot \lfloor \varepsilon \cdot N/N_{e}\rfloor ,\;$$
in which ε is a scalar within the range 1.1 to 3 [51], N is the length of signal to be analyzed, $N_{e}$ denotes the number of local extremes (including both local maxima and local minima) in the analyzed signal, and the symbol ⌊⋅⌋ represents the floor operator that rounds the value to the nearest integer toward the direction of minus infinity. We have evaluated the performance on RR estimation at different values of ε and found that 1.8 provided a reliable RR estimation for the signals that we analyzed in this research. For this reason, the value of ε is selected to be 1.8 for all of the computer experiments in this study. As mentioned in [35], there are several ways to find the local extremes in the signals. The second derivative test is adopted in the developed Python code [44]. If the value of $N_{e}$ is too small, it is possible to derive a Gaussian window that is longer than the signal length. In this situation, it is not necessary to continue the decomposition procedure. Therefore, one simple criterion that allows for further decomposition procedure is given by:(6)M<N/2−1. 

With the above background, the procedure of GAFD can be summarized as follows. 

Step 1: From the number of extremes in the signal to be decomposed, determine the value of M according to Equation (5), and then check the value of M according to Equation (6). If it is satisfied, then start or continue the decomposition procedure and generate the Gaussian window $w_{G}[\cdot ]$ of length (2M+1) based on Equations (1) and (2). Otherwise, terminate the procedure. 

Step 2: Extend the signal over both boundary points and derive the instantaneous mean $m_{i}[\cdot ]$ by conducting the moving-average operation according to Equation (3). 

Step 3: Obtain the prototype IMF r[⋅] according to Equation (4) and check whether any stop criterion has been reached. If not, repeat computing steps 1 to 3. Otherwise, stop the whole procedure. 

The second step for HGT is taking the Hilbert transform on the IMFs decomposed by GAFD. Let the i-th IMF be denoted as $x_{i}[\cdot ]$, its corresponding Hilbert transform is represented by $y_{i}[\cdot ]$. From $x_{i}[\cdot ]$ and $y_{i}[\cdot ]$, the analytic signal can be formulated as follows:(7)$$z_{i}\left[n\right]=x_{i}\left[n\right]+j\cdot y_{i}\left[n\right]=a_{i}\left[n\right]\cdot e^{j\cdot \theta_{i}\left[n\right]},\;\mathrm{for}\;0\leq n\leq N-1,\;$$
where $a_{i}[\cdot ]={\left(x_{i}^{2}[\cdot ]+y_{i}^{2}[\cdot ]\right)}^{1/2}$ and $\theta_{i}[\cdot ]=\tan^{-1}\left(y_{i}[\cdot ]/x_{i}[\cdot ]\right)$. The instantaneous frequency $\omega_{i}[\cdot ]$ is then derived from the gradient of unwrapped phase $\theta_{i}[\cdot ]$, both are in unit radian. The instantaneous frequency in unit Hz can be converted by the following equation: (8)$$f_{i}\left[n\right]=f_{s}\cdot \omega_{i}\left[n\right]/\left(2\pi \right),\;\mathrm{for}\;0\leq n\leq N-1,\;$$
in which $f_{s}$ is the sampling frequency (also in unit Hz) of the discrete-time signal. 

With the representation of the analytical signal as shown in Equation (7) and the conversion to unit Hz for instantaneous frequency according to Equation (8), each time index n now has its corresponding instantaneous frequency $f_{i}\left[n\right]$ and its corresponding amplitude $a_{i}\left[n\right]$ (or energy $a_{i}^{2}\left[n\right]$). Let $H_{i}{(n,\;f}_{i}\left[n]\right)$ denote the time-frequency-amplitude (or time-frequency-energy) representation acquired from the i-th IMF and assume that there are, in total, P IMFs derived by GAFD, the final HGT is summarized to be:(9)$$H\left(n,\;f\right)=\sum_{i=1}^{P}H_{i}{(n,\;f}_{i}\left[n]\right),\;\mathrm{for}\;0\leq n\leq N-1.\;$$

That is, the result of HGT is the sum of time-frequency-amplitude (or time-frequency-energy) representation from each IMF. As with the terminology used in HHT, such representation is called the Hilbert spectrum.

#### 2.2.2. RR Estimation Algorithm

The result of HGT is a mixture of Hilbert spectrum contributed from each IMF. To estimate RR from the mixed spectrum, the mean and the standard deviation of the instantaneous frequency are adopted in the estimation.

For instantaneous frequency $f_{i}[\cdot ]$ of the i-th IMF, the mean frequency (denoted as $\bar{f_{i}})$ and the corresponding standard deviation (denoted as $f_{i\_std}$) can be derived as follows:(10)$$\{\begin{array}{c} \bar{f_{i}}=(\overset{N-1}{\underset{n=0}{\sum }}f_{i}\left[n]\right)/N\;\\ f_{i\_\mathrm{std}}=\sqrt{\left[\overset{N-1}{\underset{n=0}{\sum }}{{(f}_{i}\left[n\right]-\bar{f_{i}})}^{2}\right]/(N-1)} \end{array}.$$

Because the rate of respiration for human beings is located within a specified range [36], the attention can be focused on this range for the derived Hilbert spectrum. Let the lower bound and upper bound of respiration frequency be denoted as $f_{rL}$ and $f_{rU}$, respectively. Assume there are, in total, P IMFs after GAFD, and there are Q possible candidates for respiration component in these IMFs. The procedure of the RR estimation algorithm is briefly summarized as follows:

Step 1: Find the possible candidates of respiration component in P IMFs by screening from the derived $\bar{f_{i}}$s that are within the range between $f_{rL}$ and $f_{rU}$. That is,
(11)$$\left\{Q\;\mathrm{IMFs}\right\}=\left\{f_{rL}\leq \bar{f_{i}}\leq f_{rU}\;:\;i=1,\;2,\;\cdots \cdots \;P\right\},$$
where {⋅} represents the symbol of a set. If Q=1 (i.e., only one IMF satisfies the requirement), then the rate of respiration $\mathrm{RR}=60\cdot \bar{f_{i}}$ (unit: bpm). If there is more than one IMF in the set, then go to Step 2.

Step 2: The respiration frequency is the mean frequency $\bar{f_{i}}$ with the smallest standard deviation among Q IMFs. That is,
(12)$$\bar{f_{i}}=\arg\underset{i}{\min}{(f}_{i\_std}\;\mathrm{in}\;\left\{Q\;\mathrm{IMFs}\right\}).$$

After respiration frequency $\bar{f_{i}}$ is determined, the rate of respiration can be estimated by $\mathrm{RR}=60\cdot \bar{f_{i}}$ (unit: bpm).

In this research, the lower bound and upper bound for respiration frequency are defined to be 0.09 Hz (i.e., 5.4 bpm) and 0.35 Hz (i.e., 21 bpm), respectively. Since no RR value in the datasets of this study exceed 21 bpm (i.e., 0.35 Hz), we selected 0.35 Hz as the upper bound for the estimation. In practical applications, the range of respiration frequency can be adjusted to meet the requirements for RR estimation. Another issue for RR estimation by the proposed method is the required signal length. It is necessary to receive the signal covered at least one cycle of respiration even under the slowest respiration frequency. For example, as the lower bound of respiration frequency is 0.09 Hz, the length of signal should be 11.11 s at least. To make the analysis more stable, a length of more than 20 s is suggested.

To evaluate the effectiveness of the proposed method, the results derived by power spectrum analysis based on complex Morlet wavelet for respiration signal are adopted for performance comparison. The detailed description on the wavelet-based spectrum analysis method may refer to [13]. In brief, let the derived spectrogram (or termed as scalogram, which is the time-frequency-energy representation acquired from wavelet transform) be denoted as $S_{c}\left(n,\;f\right)$, for 0≤n≤N−1 and $f_{rL}\leq f\leq f_{rU}$, then the time-averaged spectrum is given as follows:(13)$${\hat{s}}_{c}(f)=[\sum_{n=0}^{N-1}S_{c}\left(n,\;f\right)]/N.$$

The respiration frequency is the local maximum of ${\hat{s}}_{c}(f)$ in the interested frequency range, and RR can be computed by:(14)$$\mathrm{RR}=60\cdot \arg\underset{f_{rL}\leq f\leq f_{rU}}{\max}\left\{{\hat{s}}_{c}(f)\right\}.$$

In this research, the frequency resolution used for the computation of spectrogram is 0.01 Hz.

#### 2.2.3. Statistical Analysis

Two statistical tests were conducted in this study. The first one is the intraclass correlation coefficient (ICC) [37,38], and the other is the Bland–Altman agreement analysis [39]. ICC is used to evaluate the degree of consistency between RRs from PPG or SCG signals and RRs from respiration signal. In this research, the MATLAB code [49] that is developed based on McGraw and Wong’s article [38] is adopted in the statistical test. The two-way mixed and single score (“A-1” type in the code [49]) under 0.05 level of significance is adopted to estimate ICC. Bland–Altman analysis is utilized to evaluate whether the RRs derived by the proposed method agree well with the RRs acquired by the conventional standard approach. The limits of agreement with 95% confidence interval are used to check the scattering characteristics between the results.

## 3. Results and Discussion

There are three kinds of datasets used in the computer experiments, which are finger PPG (selected from MIMIC database [40] of PhysioNet [41]), wrist PPG (collected in Feng Chia University) and SCG (selected from CEBS database [43] of PhysioNet [41]) datasets. A detailed description of these datasets is provided in Section 2.1.1.

In this study, we propose GAFD as a signal decomposition method. To evaluate its performance, we compare it with two conventional methods, EMD and EEMD. We also test the robustness of each method under noise-corrupted conditions, with a signal-to-noise ratio (SNR) of 5 dB. Figure 1 demonstrates the results for six cases: (a) finger PPG, (b) wrist PPG, (c) SCG, (d) finger PPG with an SNR of 5 dB, (e) wrist PPG with an SNR of 5 dB and (f) SCG with an SNR of 5 dB.

The data numbered 03900006 from the MIMIC database [40] of PhysioNet [41] are adopted for the illustration in Figure 1a. From the results, the respiration component decomposed by GAFD (the 4th IMF) agrees well with respiration signal; both are evaluated to be 13 bpm by visual inspection (which has been checked and verified by the clinical expert). The values are 10 and 13 bpm for EMD (the 5th IMF) and EEMD (the 6th IMF), respectively, which were found as well from visual inspection (which has been checked and verified by the clinical expert). In addition, the boundary effect is apparent in EMD (at left end point). Such an effect still exists in EEMD (at right end point), but is not so obvious as in EMD. Figure 1b is the result of wrist PPG from one male subject aged 23, measured in the sitting posture and with a paced respiration of 6 bpm in the first minute, and changed to 10 bpm in the second minute during the experiment. It can be observed that the result decomposed by GAFD for the respiration component (in the 4th IMF) also matches well with the pattern shown in the respiration signal. The performance is relatively poorer for both EMD (the 6th IMF) and EEMD (the 6th IMF), especially at the first 10 s (where the respiration pattern is not obvious but the rugged shape may overestimate the respiration frequency). Figure 1c shows the results for the data numbered b017 from the CEBS database [43] of PhysioNet [41]. As with the other cases, the respiration component decomposed by GAFD (the 5th IMF) agrees very well with the practical respiration signal; both are nine breaths in the 30 s period. However, the results are underestimated by both EMD (the 7th IMF) and EEMD (the 7th IMF), where only eight breaths are observed in the same period. The boundary effect at the right end points looks apparent in both EMD and EEMD, and this effect influences the estimated RR. Subfigures (d)–(f) of Figure 1 illustrate the results of noise-corrupted tests on the signals presented in subfigures (a)–(c), respectively. To conduct these tests, additive white Gaussian noise (AWGN) was added to the raw signals, while maintaining a 5 dB SNR. The resulting noise-corrupted signals were then used to produce subfigures (d)–(f). The differences between subfigures (a) versus (d), (b) versus (e) and (c) versus (f) in Figure 1 can be examined to assess the noise robustness of the three methods (i.e., EMD, EEMD and GAFD). The results show that even at a low SNR of 5 dB, the proposed GAFD method is more robust to noise than the traditional methods EMD and EEMD. This finding suggests that GAFD may be a promising method for future applications in noise-robust signal processing. Figure 1 demonstrates the feasibility of GAFD in extracting the respiration component from PPG and SCG signals, even under the noise-corrupted conditions.

In the proposed method, the Hilbert transform is conducted after the IMF’s decomposition (refer to Section 2.2.1). The rate of respiration is then estimated according to the evaluation on mean and standard deviation of the instantaneous frequency from the acquired Hilbert spectrum (refer to Section 2.2.2). Figure 2, Figure 3 and Figure 4 shows the Hilbert spectrum for the analysis of finger PPG, wrist PPG and SCG, respectively. The results for the specified signal based on EMD, EEMD and GAFD are arranged in the same figure so that the comparison could be clearer and easier. In addition, the right subfigures in Figure 2, Figure 3 and Figure 4 are the Hilbert spectra sifted from the respiration-related IMF according to the proposed rule presented in Section 2.2.2.

Figure 2 is the corresponding Hilbert spectrum for the finger PPG signal shown in Figure 1a. The results based on EMD, EEMD and GAFD are shown in (a)–(c), respectively, in which the ground truth respiration frequency (dash-dotted line) and the estimated respiration frequency (dotted) are also depicted in each right subfigure. The results verify that the proposed mean and standard deviation rule (refer to Section 2.2.2) can sift the respiration-related Hilbert spectrum from the original Hilbert spectrum for all decomposition approaches. However, the error based on EMD is the largest among three approaches, which underestimates the respiration frequency just like the situation in the time domain (refer to Figure 1a).

Figure 3 depicts the corresponding Hilbert spectrum of the wrist PPG signal shown in Figure 1b. Subfigures (a)–(c) are the results by EMD, EEMD and GAFD, respectively, in which the ground truth respiration frequency (dash-dotted line) and the estimated respiration frequency (dotted) are also demonstrated in each right subfigure. From the results, it can be observed that the proposed rule (refer to Section 2.2.2) can sift the respiration-related Hilbert spectrum from the Hilbert spectrum of all IMFs in the 10-bpm cases, no matter which decomposition approach is adopted. However, the error based on EMD and EEMD overestimate the respiration frequency due to the bumpy pattern at the first 10 s (refer to Figure 1b). It can be distinguished that the estimation performance based on GAFD is still the best one.

Figure 4 is the corresponding Hilbert spectrum of the SCG signal shown in Figure 1c. The results based on EMD, EEMD and GAFD are demonstrated in (a)–(c), respectively, in which the ground truth respiration frequency (dash-dotted line) and the estimated respiration frequency (dotted) are also illustrated in each right subfigure. The results show that the proposed mean and standard deviation rule (refer to Section 2.2.2) can sift the respiration-related Hilbert spectrum from the original Hilbert spectrum for all methods. It can be appreciated that the deviation from the ground truth that is based on GAFD is the smallest one among three approaches. In this case, the error of RR is 0.37 bpm (multiply the frequency deviation by 60).

To verify the feasibility of the proposed method for RR estimation, two statistical tests were conducted in this study. There are 1000 finger PPG datasets of 60 s length (from 50 records of MIMIC database [40]), 25 wrist PPG datasets of 120 s length (from 5 healthy subjects recruited from Feng Chia University) and 500 SCG datasets of 60 s length (from 13 subjects of CEBS database [43]) covered in the test database. The RRs estimated by wavelet transform on the synchronous respiration signals are taken as the ground truth values for statistical tests. A data length of 20 s is selected for each RR estimation by the proposed method. There is no overlap for the signal segmentation of each dataset; therefore, there are a total of 4650 pairs of data for statistical tests.

The two-way mixed and single score ICC under 0.05 level of significance is conducted to check the degree of consistency between the proposed and the traditional methods. The estimated ICC is equal to 0.9688, the lower and upper bounds of ICC at 0.05 level of significance are 0.9669 and 0.9706, respectively. According to the guideline proposed by Koo and Li [52], such ICC value represents the “excellent” agreement for the RR estimation between the proposed method and the conventional approach.

Another statistical test is the Bland–Altman agreement analysis [39]. The limits of agreement with 95% confidence interval are also utilized to evaluate the scattering characteristics between the results. The result of the Bland–Altman plot is shown in Figure 5. The Pearson’s correlation coefficient (CC) is 0.9689, which is very close to the derived ICC. In addition, the mean of the bias and the limits of agreement (95% confidence interval) are also depicted in Figure 5, which is nearly 0 and around ±1 bpm, respectively. The values are low enough to show that the RRs derived by the proposed method agree well with the RRs acquired by the conventional approach.

To assess whether the proposed method exhibits bias in estimating RR from different types of signals, we performed separate computations of ICC and the Bland–Altman agreement analysis. The results of our analysis are summarized in Table 2, which indicates that there was no observable bias in estimating RR from any of the analyzed signals. Our findings suggest that the proposed method is not subject to any bias in RR estimation, as confirmed by the ICC and Bland–Altman agreement analysis.

## 4. Conclusions

This paper proposes a novel frequency estimation scheme named Hilbert–Gauss transform (HGT), which combines Gaussian average filtering decomposition (GAFD) and Hilbert transform. The principle of HGT is introduced with comprehensive mathematical formulation. In addition to being an alternative to Hilbert–Huang transform (HHT), HGT can also be used to estimate the frequency of specific components buried in the physiological signals. In this article, the examples of finger PPG, wrist PPG and SCG are demonstrated to show the feasibility of the estimation of respiratory rate (RR) based on HGT. The computer experimental results by the proposed method have been compared with those acquired by EMD- and EEMD-based approaches in different scenarios.

The decomposition performance is compared in Figure 1 for (a) finger PPG, (b) wrist PPG, (c) SCG, (d) finger PPG with an SNR of 5 dB, (e) wrist PPG with an SNR of 5 dB and (f) SCG with an SNR of 5 dB, respectively. It can be observed that GAFD can indeed extract the respiration component buried in PPG or SCG, and its performance is the most prominent among all the methods, even at a low SNR of 5 dB. For finger PPG (refer to Figure 1a,d), the boundary effect is very apparent in EMD (at left end point) and relatively mild in EEMD (at right end point). In addition, the rate of respiration may also be underestimated by EMD. In the case of wrist PPG (refer to Figure 1b,e), the respiration component decomposed by GAFD matches well with the pattern of respiration signal, even at a different rate of respiration. The performance is relatively poorer for both EMD and EEMD, especially at the first 10 s (in which the respiration pattern is not apparent and the bumpy shape leads to an overestimate on the respiration frequency). In the case of SCG decomposition (refer to Figure 1c,f), the respiration component decomposed by GAFD agrees well with the practical respiration signal. However, the rate of respiration is underestimated by both EMD and EEMD. The boundary effect at the right end points is apparent in both EMD and EEMD.

The corresponding Hilbert spectrum of finger PPG, wrist PPG and SCG are shown in Figure 2, Figure 3 and Figure 4, where the results for the specified signal based on EMD, EEMD and GAFD are organized in the same figure, and the ground truth respiration frequency and the estimated respiration frequency are also depicted in each right subfigure. The results shown in these figures verify that the proposed mean and standard deviation rule (refer to Section 2.2.2) can indeed sift the respiration-related Hilbert spectrum from the original Hilbert spectrum in each decomposition approach. In the case of finger PPG (refer to Figure 2), the error based on EMD is the largest of all approaches, where the respiration frequency has been underestimated (just like the phenomenon observed in the time domain). For wrist PPG (refer to Figure 3), the respiration frequency is overestimated in both EMD- and EEMD-based approaches for the first-minute period because of the rugged shape at the first 10 s. In the example of SCG (refer to Figure 4), the deviation from the ground truth that is based on GAFD is the smallest one in all approaches. In this demonstrated case, the error of RR is 0.37 bpm.

There are a total of 4650 pairs of data (the RRs estimated by the proposed method from PPG or SCG, and the RRs estimated by wavelet transform from respiration signal) selected for statistical tests from three different databases. The estimated ICC is 0.9688, and the lower and upper bounds of ICC (0.05 level of significance) are 0.9669 and 0.9706, respectively. It is evaluated to be in “excellent” agreement (according to the guideline proposed in [52]) for the RR estimation between the proposed method and the traditional approach. In addition, Bland–Altman’s limits of agreement (95% confidence interval) are derived to be around ±1 bpm (refer to Figure 5), respectively. The values are low enough to show that the RRs derived by the proposed method agree well with the RRs acquired by the conventional approach. With the separate analyses in estimating RR from different types of signals, it can be observed there was no observable bias for the examined signals (refer to Table 2). The results suggest that the proposed method is not subject to any bias in RR estimation.

The demonstrated results verified that the proposed HGT-based approach can be used to estimate RR from PPG or SCG signal reliably, even in the respiration pace-changed conditions. This paper shows the superior performance of GAFD compared to EMD and EEMD in certain examples. In addition to biomedical applications, EMD- or EEMD-based HHT was applied in many scenarios. The proposed HGT has the potential to play a similar role as HHT has previously.

## Figures and Tables

**Figure 1 sensors-23-03785-f001:**
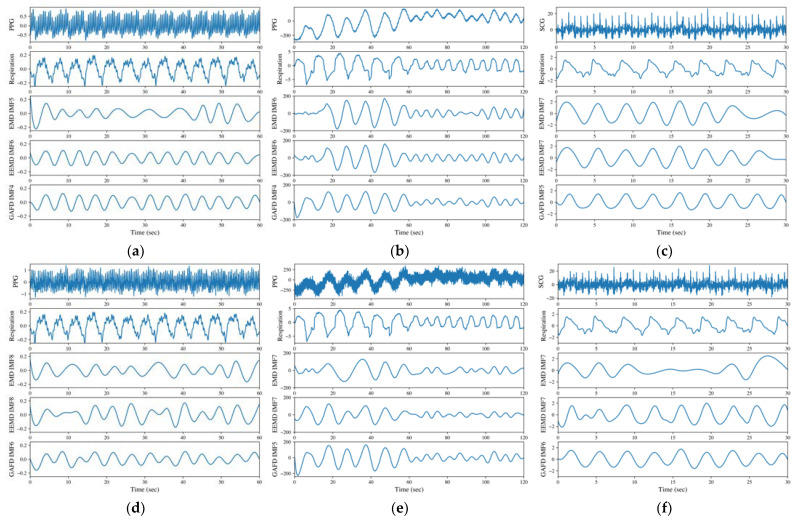
Decomposition results for (**a**) finger PPG, (**b**) wrist PPG, (**c**) SCG, (**d**) finger PPG with an SNR of 5 dB, (**e**) wrist PPG with an SNR of 5 dB and (**f**) SCG with an SNR of 5 dB. The illustrations are arranged in order of signal type (PPG or SCG), respiration signal and the respiration-related IMF decomposed by EMD, EEMD and GAFD (from top to bottom).

**Figure 2 sensors-23-03785-f002:**
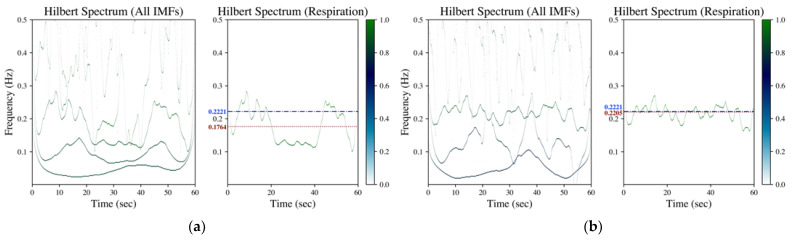
Hilbert spectrum of finger PPG (data numbered 03900006 from MIMIC database) by (**a**) EMD, (**b**) EEMD and (**c**) GAFD. The illustrations in each subfigure are the Hilbert spectrum for all IMFs on the left, whereas the Hilbert spectrum for the respiration-related IMF is on the right. In the Hilbert spectrum of respiration (right subfigure), the dash-dotted line (blue) denotes the respiration frequency estimated by wavelet transform from respiration signal (ground truth), whereas the dotted line (dark red) represents the respiration frequency estimated from the Hilbert spectrum according to the rule introduced in Section 2.2.2.

**Figure 3 sensors-23-03785-f003:**
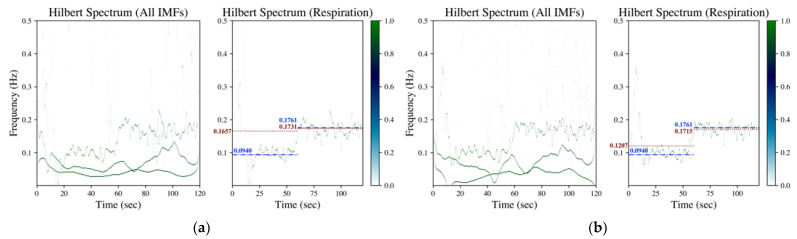
Hilbert spectrum of wrist PPG (measured from a male subject aged 23 in the sitting posture with a controlled respiration of 6 bpm in the first minute and changed to 10 bpm in the second minute) by (**a**) EMD, (**b**) EEMD and (**c**) GAFD. The illustrations in each subfigure are the Hilbert spectrum for all IMFs on the left, whereas the Hilbert spectrum for the respiration-related IMF is on the right. In the Hilbert spectrum of respiration (right subfigure), the dash-dotted line (blue) denotes the respiration frequency estimated by wavelet transform from respiration signal (ground truth), whereas the dotted line (dark red) represents the respiration frequency estimated from the Hilbert spectrum according to the rule introduced in Section 2.2.2.

**Figure 4 sensors-23-03785-f004:**
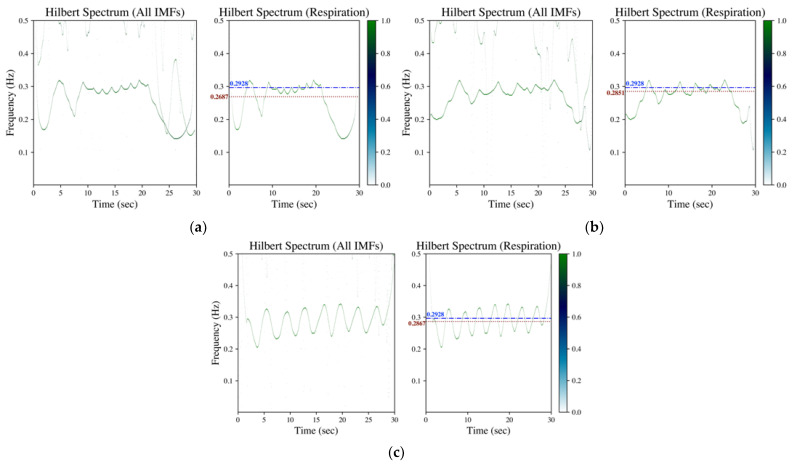
Hilbert spectrum of SCG (data numbered b017 from CEBS database) by (**a**) EMD, (**b**) EEMD and (**c**) GAFD. The illustrations in each subfigure are the Hilbert spectrum for all IMFs on the left, whereas the Hilbert spectrum for the respiration-related IMF is on the right. In the Hilbert spectrum of respiration (right subfigure), the dash-dotted line (blue) denotes the respiration frequency estimated by wavelet transform from respiration signal (ground truth), whereas the dotted line (dark red) represents the respiration frequency estimated from the Hilbert spectrum according to the rule introduced in Section 2.2.2.

**Figure 5 sensors-23-03785-f005:**
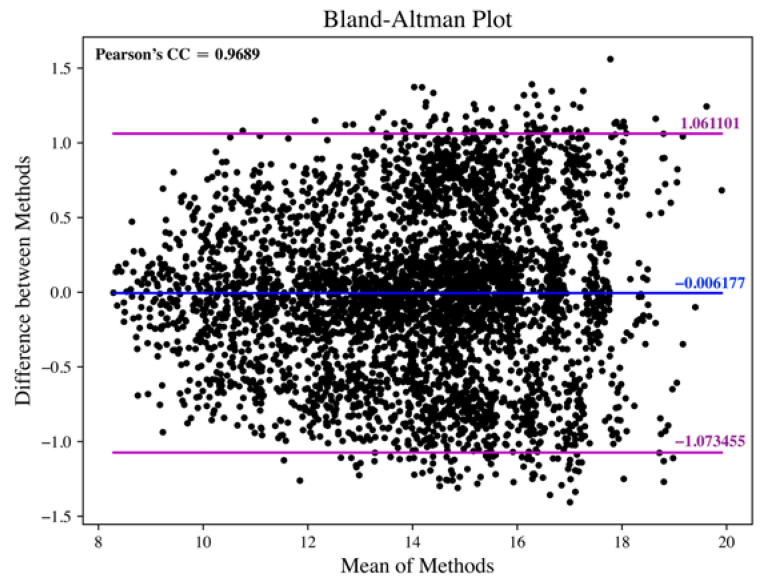
Bland–Altman plot for the RR estimation between the proposed method and the conventional approach.

**Table 1 sensors-23-03785-t001:** Summary of the datasets used in this research.

	Subject Number ^1^	Age Range ^2^	Posture	Record Number ^3^
Finger PPG	50 (30M, 20F)	21–92 years	Supine (ICU)	3000 records
Wrist PPG	5 (3M, 2F)	22–24 years	Sitting	150 records
SCG	13 (6M, 7F)	19–30 years	Supine	1500 records

**^1^** Subject Number indicates the total number of subjects included in the dataset, with ‘M’ representing male and F representing female. ^2^ Age Range refers to the range of age range for the participated subjects. ^3^ Record Number is the number of records of 20 s length in the dataset.

**Table 2 sensors-23-03785-t002:** Summary of ICC and Bland–Altman agreement analysis for RR estimation under different signal conditions.

	ICC ^1^	95% C.I. of ICC ^2^	Bias ^3^	Limit of Agreement ^4^
Finger PPG	0.9680	0.9655, 0.9704	−0.0048±0.5493	−1.1033, 1.0937
Wrist PPG	0.9650	0.9573, 0.9714	0.0018±0.5315	−1.0611, 1.0647
SCG	0.9691	0.9658, 0.9720	−0.0104±0.5403	−1.0910, 1.0702
Combined **^5^**	0.9688	0.9669, 0.9706	−0.0062±0.5446	−1.0954, 1.0830

^1^ ICC means the intraclass correlation coefficient.^2^ C.I. is the abbreviation of confidence interval. ^3^ Bias denotes the difference between the estimated and the ground truth values, which is presented as mean ± SD (standard deviation) in the unit of bpm. ^4^ Limit of Agreement represents the lower and the upper limit of agreement for the Bland–Altman agreement analysis between the estimated values and the ground truth values derived by the conventional approach in the unit of bpm. ^5^ “Combined” refers to the fusion of finger PPG, wrist PPG and SCG signals for ICC and Bland–Altman agreement analysis.

## Data Availability

The data used in this study are available upon request.

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
