# Peer review of "A Frequency Estimation Scheme Based on Gaussian Average Filtering Decomposition and Hilbert Transform: With Estimation of Respiratory Rate as an Example"

_sensors, 2023, doi:10.3390/s23083785_

Round 1
Reviewer 1 Report
This article proposes a new frequency estimation scheme for the Hilbert-Gauss transform (HGT), which combines the Gaussian average filtering decomposition (GAFD) and the Hilbert transform. The principles of HGT are also introduced. In addition to serving as a replacement for the Hilbert-Huang transform (HHT), HGT can also be used to estimate the frequency of specific components buried in physiological signals.
The article uses finger PPG, wrist PPG, and SCG as examples to demonstrate the feasibility of using HGT to estimate respiratory rate (RR). The experimental results of this method are compared with those of the basic EMD and EEMD methods in different scenarios, and the results show that even when the respiratory step speed changes, the proposed HGT-based method can reliably estimate the RR of PPG or SCG signals.
However, there are still some problems with this article. For example, in terms of formatting, there are many problems with the numbering of formulas. The comparison of experimental data in the article is presented in the form of images, which could be more intuitive if summarized in a table for comparison.
Author Response
Thank you for taking the time to review our manuscript. We greatly appreciate your constructive feedback and suggestions for improvement. We have addressed the equation formatting issue you mentioned according to the standard template of Sensors (refer to Equations (1) to (14) from page 5 to page 8). For your concern on the comparison of experimental data, we have added a new table (Table 2) titled "Summary of ICC and Bland-Altman’s agreement analysis for RR estimation under different signal conditions" (lines 521-531 on page 14) in the revision. This new table presents the results of our separate computations of ICC and Bland-Altman's agreement analysis for each of the three types of signals used in our study. Table 2 shows that there was no observable bias in estimating RR from any of the analyzed signals. Therefore, our findings suggest that the proposed method is not subject to any bias in RR estimation, which is supported by both the ICC and Bland-Altman agreement analysis. In addition, we have incorporated your suggestion into our Conclusion section by stating that "with the separate analyses in estimating RR from different types of signals, it can be observed that there was no observable bias for the examined signals (refer to Table 2). The results suggest that the proposed method is not subject to any bias in RR estimation" (lines 581-584 on page 15).
The revised and newly added portion has been highlighted in blue color for the revised manuscript. Thank you again for your valuable feedback and for the time and effort you put into reviewing our manuscript. We hope our revisions have addressed your concerns and meet the standards of Sensors.
Reviewer 2 Report
I have reviewed the manuscript. It is a commendable work. I have not found any methodological flaws. The result is well presented. This paper match with the SI content.
I have not any negative comments and therefore I recommend to accept this paper in present form.
Thank you
Author Response
Thank you for taking the time to review our manuscript. We greatly appreciate your positive feedback and your assessment that our work is commendable. With your kind help, we are very glad and feel honored to possibly publish a highly qualified research paper in Sensors.
We are grateful to hear that you recommend accepting the paper in its present form. We are willing to address any further concerns or suggestions you may have, and we look forward to the opportunity to improve our manuscript further with your guidance.
Once again, thank you for your valuable feedback and for the time and effort you put into reviewing our manuscript.Reviewer 3 Report
It is a very interesting work with important applications in time-frequency analysis.
You have proposed and measured the performance of a novel and interesting combination of techniques for extracting respiratory frequency using databases of subjects at rest, and in cases where possible movement artifacts or signal alterations were detected, you have eliminated cases that could affect the performance of the technique. Please discuss the sensitivity of the proposed technique when the original signals are affected by motion artifacts, saturations, or other disturbances. This should be considered in light of the fact that the performance of the technique has been compared to cases where the same "atypical" cases were not suppressed.
If the technique sensitivity to noise or disturbances is very high, it is important to note that a weakness of the technique is that it requires signals free of motion artifacts and other disturbances, therefore it would have application in the case of sedated patients or those in surgery.
Most equations require adjustment in the placement of equation numbering, please keep this in mind.
Author Response
Thank you for taking the time to review our manuscript. We greatly appreciate your constructive feedback and suggestions for improvement.
For your concerns on the noise tolerance of the proposed method, three kinds of situations have been mentioned in your comments. For the case of motion artifacts, all of the signals (including the respiratory signal) in the databases are almost disturbed during the motion and this makes the verification a difficult task. For the case of saturation, the signals have no change in the patterns and there is no information can be provided for the analysis on respiration activity. To attain the goal of noise robustness, the key point is on whether the respiratory pattern can be decomposed from the noise-corrupted signal. We have thus used the artificially generated additive white Gaussian noise (AWGN) to raw signals and compared the decomposing results by EMD, EEMD and the proposed GAFD, respectively. The results have been incorporated into Figure 1 with subfigures (d), (e) and (f) (lines 380-381 on page 9), and we have added the related content as follows in the context (lines 408-419 on page 10):
“Subfigures (d), (e) and (f) of Figure 1 illustrate the results of noise-corrupted tests on the signals presented in subfigures (a), (b) and (c), respectively. To conduct these tests, additive white Gaussian noise (AWGN) was added to the raw signals, while maintaining a 5 dB SNR. The resulting noise-corrupted signals were then used to produce subfigures (d), (e) and (f). The differences between subfigures (a) versus (d), (b) versus (e), and (c) versus (f) in Figure 1 can be examined to assess the noise robustness of the three methods (i.e., EMD, EEMD and GAFD). The results show that, even at a low SNR of 5 dB, the proposed GAFD method is more robust to noise than the traditional methods EMD and EEMD. This finding suggests that GAFD may be a promising method for future applications in noise-robust signal processing. Figure 1 demonstrates the feasibility of GAFD in extracting the respiration component from PPG and SCG signals, even under the noise-corrupted conditions.”
We have addressed the adjustment in the placement of equation numbering you mentioned according to the standard template of Sensors (refer to Equations (1) to (14) from page 5 to page 8).
The revised and newly added portion has been highlighted in blue color for the revised manuscr ipt. We have carefully considered your suggestions and made significant revisions to address the issues raised in your comment. Thank you again for your valuable feedback and for the time and effort you put into reviewing our manuscript.Reviewer 4 Report
The paper presents an interesting approach for estimating a respiratory rate by combining Gaussian decomposition and Hilbert-Huang transform. The proposed approach may useful for different clinical devises monitoring respiratory rate as well as for custom electronics. Although the paper is well organized and clearly written, I have several concerns and comments for the authors, that are summarized below. I hope that they will help to improve the manuscript.
Comments:
1. Subsection 2.1.1. For 50 record utilized for the research from MIMIC the information about subjects for these data is missing. Also it is not clear how many segments (or percentage) from 50 records were not selected for further analysis while using LightWAVE, which level of interference was taken as a threshold exclude a segment from further consideration.
2. A table describing the key parameters of all three used databases would be useful for comparing them: by the number and gender of the subjects, the conditions of the experiment, the weight in the combined database on which the proposed method was tested.
3. P.5, Formulas and formula numbers need proper formatting
4. P.5, lines 241-242 ‘This paper adopts “double-symmetrical reflection” for all computer experiments.’ Why this extension style has been chosen by authors?
5. P.6, line 274. It is no clear how authors picked ε = 1.8. Was any kind of analysis used to select it as optimal value?
6. P. 7 line 340. As it is known from physiology a respiratory rate may excide 0.35 Hz at rest state especially if we a dealing with persons with pulmonary health problem or children. Why maximum possible respiration frequency was limited by authors by 0.35 Hz?
7. Lines in Figures 2-4 are hardly readable, their quality needs improvement.
8. P.9 authors stated that IMFs for EMD and EEMD were chosen ‘from visual inspection’. It is unclear if the operator while such visual inspection had any a prior knowledge about RR ground truth for the record analyzed or not.
9. As 3 datasets used in the paper different in methods (finger PPT, wrist PPT, and SCG), it would be worth providing a table with an assessment of the effectiveness of the proposed method, depending on the dataset.
Author Response
Comments:
1.Subsection 2.1.1. For 50 records utilized for the research from MIMIC the information about subjects for these data is missing. Also it is not clear how many segments (or percentage) from 50 records were not selected for further analysis while using LightWAVE, which level of interference was taken as a threshold exclude a segment from further consideration.
Reply:
Thank you for providing feedback on our manuscript. We appreciate your constructive comments and suggestions for improvement. We have made the necessary changes to the manuscript based on your suggestions.
Regarding your concerns, we used 50 records from the MIMIC dataset for our research. Although information about subjects was missing, we have added a summary of the relevant information in Table 1 (lines 208-212 on page 5) for the convenience of the reader. For more detailed information, we have also included the MIMIC website in the context (lines 148-149 on page 3).
To select the 1,000 segments, we used the online visual tool LightWAVE to prescreen the signal quality and mark the segments of good quality. We collaborated with Dr. Tienhsiung Ku, a clinical expert and one of the authors, to ensure that the selected segments met the criterion of good signal quality to avoid any misleading results in the further analysis. For this issue, we have added the content “The signal selection and the experiments were conducted under the supervision of clinical expert in our team to assure the quality of signals collected” (lines 203-204 on page 4) in the context of the revised manuscript.
We selected 20 segments from each of the 50 records, with a total of 1,000 segments being chosen. Additionally, we clarified that each segment is one-minute long, and each segment was separated into three 20-second shorter segments for RR estimation.
Thank you again for your valuable feedback and for the time and effort you put into reviewing our manuscript. We hope our revisions have addressed your concerns.
- A table describing the key parameters of all three used databases would be useful for comparing them: by the number and gender of the subjects, the conditions of the experiment, the weight in the combined database on which the proposed method was tested.
Reply:
Thank you for taking the time to review our manuscript. We greatly appreciate your constructive feedback and suggestions for improvement. We have incorporated a new table (Table 1, at lines 208-212 on page 5) summarizing the key parameters of the datasets used in this research to address the reviewer's comment on the need for a comparison of the datasets. The table provides information on the number of subjects, age range, experimental conditions, and the number of records in each dataset. We hope that this will help the reader better understand the datasets used in this study and facilitate comparison across them.
Thank you again for your valuable feedback and for the time and effort you put into reviewing our manuscript. We hope our revisions have addressed your concerns.
- P.5, Formulas and formula numbers need proper formatting.
Reply:
Thank you for taking the time to review our manuscript. We greatly appreciate your constructive feedback and suggestions for improvement. We have addressed the equation formatting issue you mentioned according to the standard template of Sensors (refer to Equations (1) to (14) from page 5 to page 8). The revised portion has been highlighted in blue color for the revised manuscript.
Thank you again for your valuable feedback and for the time and effort you put into reviewing our manuscript. We hope our revisions have addressed your concerns and meet the standards of Sensors.
- P.5, lines 241-242 ‘This paper adopts “double-symmetrical reflection” for all computer experiments.’ Why this extension style has been chosen by authors?
Reply:
Thank you your valuable feedback on our manuscript. Regarding the question about the choice of "double-symmetrical reflection" for computer experiments in our paper, we have adopted this extension style because we found it to be a feasible choice for eliminating the boundary effect in signal decomposition. Our team has tested various signal types, and this method proved to be effective. We believe that this is because the reflection mode can make the signal reflect more smoothly on the boundary edge. Although it is not easy to verify mathematically, we mention this extension method in our research based on our experimental findings. We have addressed this issue in the context by “because this extension style can effectively eliminate the boundary effect during the decomposition procedure based on the experimental findings” at lines 250-251 on page 6.
We hope this clarification addresses your concern and thank you again for your valuable feedback and for the time and effort you put into reviewing our manuscript.
- P.6, line 274. It is no clear how authors picked ε = 1.8. Was any kind of analysis used to select it as optimal value?
Reply:
Thank you for your thoughtful review of our manuscript and for providing us with constructive feedback. We have carefully considered your question about our choice of ε in our computer experiments and have made the appropriate revisions to address your concern.
For your concern on the selection of ε, we would like to refer you to Reference [51] (Lin, L.; Wang, Y.; Zhou, H., Iterative filtering as an alternative algorithm for empirical mode decomposition. Advances in Adaptive Data Analysis 2009, 1, (04), 543-560.), which recommends a range of values between 1.1 and 3 for this parameter. In our research, we conducted tests to determine the optimal value of ε for our signals. Specifically, we evaluated the performance on RR estimation at different values of ε and found that 1.8 provided a reliable RR estimation for the signals that we analyzed in this research. We have added a paragraph to the manuscript (lines 277-281 on page 6) that explains how we determined the optimal value of ε for our signals and why we chose 1.8 as the value for our experiments.
We hope that this explanation satisfactorily address your inquiry. We appreciate your valuable feedback and the time you took to review our manuscript.
- P. 7 line 340. As it is known from physiology a respiratory rate may excide 0.35 Hz at rest state especially if we a dealing with persons with pulmonary health problem or children. Why maximum possible respiration frequency was limited by authors by 0.35 Hz?
Reply:
Thank you for taking the time to review our manuscript and providing specific feedback on the upper bound of respiration frequency. Since no RR value in the datasets of this study exceed 21 bpm (i.e., 0.35 Hz), we selected 0.35 Hz as the upper bound for the estimation. In practical applications, the range of respiration frequency can be adjusted to meet the requirements for RR estimation. However, in practical applications, the range of respiration frequency can be adjusted to meet the requirements for RR estimation. We have added a paragraph to the manuscript (at lines 336-339 on page 8) to address this issue. Thank you again for your valuable feedback, and for the time and effort you put into reviewing our manuscript.
- Lines in Figures 2-4 are hardly readable, their quality needs improvement.
Reply:
Thank you for taking the time to review our manuscript and providing valuable feedback. We appreciate your suggestion regarding the readability and quality of the lines in Figures 2-4 (lines 420-426 on page 10 for Figure 2, lines 448-455 on page 11 for Figure 3, and lines 469-478 on page 12 for Figure 4). We have followed your advice and made the necessary improvements by slightly thickening the lines and changing the ground truth line to blue color, which have improved the readability of Figures 2-4.
Thank you again for your valuable feedback and for the time and effort you put into reviewing our manuscript.
- P.9 authors stated that IMFs for EMD and EEMD were chosen ‘from visual inspection’. It is unclear if the operator while such visual inspection had any a prior knowledge about RR ground truth for the record analyzed or not.
Reply:
Thank you for your valuable feedback and for bringing to our attention the concern about the selection of IMFs in our manuscript. We acknowledge that the visual inspection process for IMF selection may raise questions about the operator's prior knowledge of RR ground truth for the analyzed record. To address this concern, we would like to clarify that one of the authors, Dr. Tienhsiung Ku, is a clinical doctor with expertise in biomedical signal analysis, and he carefully examined and verified the IMF selection to ensure its correctness. We have now added a statement to the manuscript to highlight this (line 390 and line 392 on page 9).
Thank you again for your valuable feedback and for the time and effort you put into reviewing our manuscript.
- As 3 datasets used in the paper different in methods (finger PPG, wrist PPG, and SCG), it would be worth providing a table with an assessment of the effectiveness of the proposed method, depending on the dataset.
Reply:
Thank you for your valuable feedback on our paper. We have carefully considered your suggestion and have made significant revisions to address the concern raised in your comment. Specifically, we have added a new table (Table 2) titled "Summary of ICC and Bland-Altman’s agreement analysis for RR estimation under different signal conditions" (lines 521-531 on page 14) in the revision. This new table presents the results of our separate computations of ICC and Bland-Altman's agreement analysis for each of the three types of signals used in our study. Table 2 shows that there was no observable bias in estimating RR from any of the analyzed signals. Therefore, our findings suggest that the proposed method is not subject to any bias in RR estimation, which is supported by both the ICC and Bland-Altman agreement analysis. In addition, we have incorporated your suggestion into our Conclusion section by stating that "with the separate analyses in estimating RR from different types of signals, it can be observed that there was no observable bias for the examined signals (refer to Table 2). The results suggest that the proposed method is not subject to any bias in RR estimation" (at lines 581-584 on page 15).
The revised and newly added portion has been highlighted in blue color for the revised manuscript. We have carefully considered your suggestions and made significant revisions to address the issues raised in your comment. Thank you again for your valuable feedback and for the time and effort you put into reviewing our manuscript.